# miR-146a Decreases Inflammation and ROS Production in Aged Dermal Fibroblasts

**DOI:** 10.3390/ijms25136821

**Published:** 2024-06-21

**Authors:** Liping Zhang, Iris C. Wang, Songmei Meng, Junwang Xu

**Affiliations:** 1Department of Physiology, College of Medicine, University of Tennessee Health Science Center, Memphis, TN 38163, USA; lzhan112@uthsc.edu (L.Z.); iris_c_wang@brown.edu (I.C.W.); smeng@uthsc.edu (S.M.); 2Division of Applied Mathematics, Brown University, Providence, RI 02912, USA

**Keywords:** aging wound healing, dermal fibroblasts, miR-146a, reactive oxygen species (ROS)

## Abstract

Aging is associated with a decline in the functionality of various cell types, including dermal fibroblasts, which play a crucial role in maintaining skin homeostasis and wound healing. Chronic inflammation and increased reactive oxygen species (ROS) production are hallmark features of aging, contributing to impaired wound healing. MicroRNA-146a (miR-146a) has been implicated as a critical regulator of inflammation and oxidative stress in different cell types, yet its role in aged dermal fibroblasts and its potential relevance to wound healing remains poorly understood. We hypothesize that miR-146a is differentially expressed in aged dermal fibroblasts and that overexpression of miR-146a will decrease aging-induced inflammatory responses and ROS production. Primary dermal fibroblasts were isolated from the skin of 17-week-old (young) and 88-week-old (aged) mice. Overexpression of miR-146a was achieved through miR-146a mimic transfection. ROS were detected using a reliable fluorogenic marker, 2,7-dichlorofluorescin diacetate. Real-time PCR was used to quantify relative gene expression. Our investigation revealed a significant reduction in miR-146a expression in aged dermal fibroblasts compared to their younger counterparts. Moreover, aged dermal fibroblasts exhibited heightened levels of inflammatory responses and increased ROS production. Importantly, the overexpression of miR-146a through miR-146a mimic transfection led to a substantial reduction in inflammatory responses through modulation of the NF-kB pathway in aged dermal fibroblasts. Additionally, the overexpression of miR-146a led to a substantial decrease in ROS production, achieved through the downregulation of NOX4 expression in aged dermal fibroblasts. These findings underscore the pivotal role of miR-146a in mitigating both inflammatory responses and ROS production in aged dermal fibroblasts, highlighting its potential as a therapeutic target for addressing age-related skin wound healing.

## 1. Introduction

The elderly population, constituting 12.4% (35 million) of the total US population, is experiencing rapid growth and is expected to reach 20% (53 million) by 2030 (U.S. Census, 2000). Among this demographic, chronic wounds like pressure ulcers, diabetic foot ulcers, and venous leg ulcers are prevalent, affecting approximately 3% of individuals aged over 65 in the United States [1]. The annual cost of caring for chronic wounds in the US is estimated at around USD 10 billion, with a significant portion likely attributed to wound care for adults aged 65 and above [2]. Given the lack of effective treatments for age-related wound healing, addressing this critical issue is imperative.

Aging is a natural process that is characterized by a gradual decline in the physiological functions of various tissues and organs. As individuals age, their ability to heal wounds becomes progressively impaired, and the healing process becomes slower and less effective [3]. This decline in wound-healing ability has been attributed to a variety of factors, including changes in the immune system, alterations in the extracellular matrix, and decreased angiogenesis [4]. In the context of skin, aging is associated with structural changes, such as thinning of the epidermis and dermis, and functional alterations in resident cells, including dermal fibroblasts [5,6]. Dermal fibroblasts are key players in skin homeostasis and wound healing, responsible for synthesizing extracellular matrix components and secreting growth factors essential for tissue repair [7,8]. However, aging-related changes in dermal fibroblasts, including impaired proliferation, reduced collagen production, and altered response to stimuli, contribute to delayed wound healing and increased susceptibility to chronic wounds in the elderly population [9,10].

Chronic inflammation and ROS are critical factors in the pathogenesis of age-related skin disorders and impaired wound healing [11,12,13,14,15]. In aging skin, dysregulated immune responses lead to an imbalance of pro-inflammatory and anti-inflammatory cytokines, resulting in a prolonged inflammatory phase that hinders timely progression to tissue repair [5,14]. This is compounded by delayed macrophage polarization and excessive matrix metalloproteinase (MMP) activity, causing tissue breakdown and fibrosis [6,16]. Concurrently, increased ROS production due to mitochondrial dysfunction and environmental factors like UV radiation damages cellular components such as keratinocytes and endothelial cells, impairing their function in wound closure and angiogenesis [12,13,17].

A key player in this process is the NF-κB signaling pathway, which is activated by ROS and further exacerbates inflammation. The sustained activation of NF-κB in aging skin leads to increased production of inflammatory cytokines and chemokines, perpetuating a cycle of chronic inflammation [18,19]. Additionally, the enzyme NADPH oxidase 4 (NOX4) has been identified as a significant source of ROS in aging dermal fibroblasts. NOX4’s elevated activity in aged cells contributes to oxidative stress by producing ROS, which damage cellular structures and amplify inflammatory signaling [17,20,21].

ROS also activate the NF-κB signaling pathway, exacerbating inflammation, while aging reduces the efficacy of antioxidant defenses, allowing cumulative oxidative damage. The combined impact of chronic inflammation and oxidative stress delays wound-healing phases, increases infection risk, and reduces collagen synthesis, leading to weaker scar tissue. Addressing these factors through targeted anti-inflammatory and antioxidant therapies, as well as innovative approaches like stem cell therapy, holds promise for improving wound care and skin health in the elderly. miR-146a is a pivotal regulator in the aging process, exerting its influence through multiple pathways [18,22,23]. miR-146a is a key modulator of immune responses, particularly in the context of chronic inflammation, a hallmark of aging-related disorders [18,22]. Beyond inflammation, miR-146a’s impact extends to tissue repair and regeneration, where it orchestrates the expression of genes crucial for maintaining tissue homeostasis and combating age-related damage [24]. Moreover, miR-146a is involved in regulating cellular senescence [25]. Indeed, miR-146a has been identified as a critical modulator of inflammatory responses and oxidative stress in various cell types, holding potential for therapeutic interventions targeting age-related disorders. Yet its role in aged dermal fibroblasts and its potential implications for age-related skin wound healing remain poorly understood. We hypothesize that miR-146a is differentially expressed in aged dermal fibroblasts and that overexpression of miR-146a will decrease aging-induced inflammatory responses and ROS production.

## 2. Results

### 2.1. Delayed Wound Healing and Dysregulated miR-146a and NOX4 Gene Expression in Aged Mouse Wounds

We conducted a comparative analysis of the wound-healing process in young and aged mice. To do so, we induced 8 mm incision wounds on the dorsal region of the mice. The experimental groups consisted of aged mice (88 weeks old, *n* = 5) and young mice (17 weeks old, *n* = 5). Following wound induction, we monitored the progression of wound closure over time. Our observations revealed a notable discrepancy in the rate of wound healing between the two groups (Figure 1A). We found that the wounds in aged mice exhibited a significantly slower rate of closure compared to those in young mice. In particular, the wounds in aged mice took approximately 18 days to fully close post-injury, as shown by the orange line in Figure 1B. In contrast, the wounds in young wild-type mice reached complete closure around day 14, as illustrated by the blue line in Figure 1B. This discrepancy in healing time between aged and young mice indicates a delay of approximately 4 days in wound closure in the aged group. These findings underscore the impact of age on the wound-healing process, suggesting that aging may adversely affect the efficiency and speed of wound repair.

We then compared the expression levels of miR-146a and NOX4 genes in wounds from aged and young mouse wounds. miR-146a, a key regulator of immune responses and oxidative stress, plays a pivotal role in the aging process and is considered a marker for inflammaging. NOX4 is the primary enzyme responsible for ROS production in skin wounds. Our findings revealed that wounds in aged mice exhibited lower miR-146a expression and higher NOX4 expression (Figure 1C,D). These results indicate that aging likely influences the regulation of inflammatory responses and ROS in the wound-healing process.

### 2.2. miR-146a and NOX4 Expression in Aged Dermal Fibroblasts

Analysis using quantitative real-time PCR demonstrated a notable decline in the expression levels of miR-146a within aged dermal fibroblasts when compared to their younger counterparts. Specifically, the expression of miR-146a exhibited a reduction of approximately 50% in aged fibroblasts, indicating a disrupted regulation of this crucial regulatory molecule during the aging process (Figure 2A). Conversely, there was a marked elevation in the expression of NOX4 within the aged fibroblasts. This increase in NOX4 expression was substantial, showing a 4-fold induction in aged fibroblasts compared to their younger counterparts (Figure 2B).

### 2.3. Inflammatory Responses and ROS Production in Aged Dermal Fibroblasts

A comparison between aged and young dermal fibroblasts revealed a pronounced increase in inflammatory responses in the aged dermal fibroblasts. This was demonstrated by the heightened expression of pro-inflammatory cytokines, including interleukin-6 (IL-6) (Figure 3A) and NF-kB (Figure 3D), along with elevated levels of IRAK1 (Figure 3B) and TRAF6 (Figure 3C), pivotal components of the NF-KB pathway and targets of miR-146a [19], as assessed through quantitative real-time PCR analysis. In particular, the expression levels of IL-6 were found to be approximately 10 times higher in aged fibroblasts compared to their young counterparts (Figure 3A), indicating an inflammatory phenotype associated with aging.

Moreover, aged dermal fibroblasts exhibited elevated levels of ROS compared to young fibroblasts. Utilizing the fluorogenic marker 2,7-dichlorofluorescin diacetate, a significant increase in ROS levels was detected in aged fibroblasts, as evidenced by intensified fluorescence intensity, with a 4.9% increase in ROS levels in aged dermal fibroblasts. (Figure 3E). This heightened ROS production aligns with the oxidative stress commonly observed in aging, which may contribute to impaired wound healing in aged skin.

Regarding morphology, our findings revealed a notable difference in size between aged and young dermal fibroblasts. Specifically, aged fibroblasts were observed to be significantly larger than their young counterparts, with a size approximately double that of young fibroblasts (Figure 3F).

### 2.4. Effects of miR-146a Overexpression on Inflammatory Responses and ROS Production

To investigate the potential involvement of miR-146a in anti-inflammatory responses and ROS production, dermal fibroblasts were transfected with either a miR-146a mimic or a control mimic. Our findings revealed a significant elevation in miR-146a RNA levels within the cells transfected with the miR-146a mimic, indicating successful transfection (Figure 4A). Transfecting aged dermal fibroblasts with a miR-146a mimic yielded a substantial decrease in inflammatory responses. Real-time PCR analysis demonstrated a significant reduction in IL-6 expression levels following miR-146a overexpression (Figure 4B), indicating suppression of the inflammatory cascade. This anti-inflammatory effect was, in part, mediated through modulation of the NF-κB pathway, a pivotal regulator of inflammatory gene expression. Moreover, overexpression of miR-146a led to significant reductions in IRAK1, TRAF6, and NF-kB levels in aged dermal fibroblasts (Figure 4C–E). Similarly, in young dermal fibroblasts, overexpression of miR-146a resulted in significantly reduced levels of IL-6, TRAF6, and NF-kB, with no significant change observed for IRAK1 (Figure 4B–E).

Furthermore, miR-146a overexpression in aged dermal fibroblasts induced a significant decrease in ROS production in both aged and young fibroblasts. Fluorescence intensity measurements revealed a marked reduction in ROS levels in miR-146a-overexpressing fibroblasts compared to control cells, with a 3.1% decrease in ROS levels observed in young fibroblasts and a 4.4% decrease in aged fibroblasts (Figure 4F).

### 2.5. miR-146a Targets NOX4 in Dermal Fibroblasts

To comprehend how miR-146a overexpression mitigates ROS production in aged dermal fibroblasts, we conducted in silico analysis, which unveiled a potential interaction between miR-146a and the 3’-UTR region of NOX4 (Figure 5A). Subsequently, we investigated whether miR-146a overexpression could influence NOX4 expression levels. Remarkably, our analysis revealed that miR-146a overexpression led to a significant reduction in NOX4 expression levels, by approximately 3-fold (Figure 5B). This finding underscores the regulatory role of miR-146a in modulating NOX4 expression, providing mechanistic insights into how miR-146a exerts its antioxidative effects in aged dermal fibroblasts. In summary, we believe that miR-146a targets key components of the NF-κB pathway and NOX4, exerting anti-inflammatory and antioxidant effects. By doing so, it helps mitigate the inflammatory and oxidative stress associated with aging, as we illustrated in Figure 5C.

## 3. Discussion

Our findings highlight the pivotal role of miR-146a in modulating inflammatory responses and oxidative stress in aged dermal fibroblasts. MiR-146a has been previously identified as a key regulator of immune responses and oxidative stress in various cell types [21,22,23,26]. Its downregulation in aged fibroblasts suggests a dysregulation of endogenous mechanisms that maintain cellular homeostasis and tissue repair capacity with aging. By targeting key components of the NF-κB pathway and NOX4, miR-146a exerts anti-inflammatory and antioxidant effects, thereby mitigating the inflammatory and oxidative burden associated with aging.

The dysregulation of inflammatory responses and oxidative stress in aged dermal fibroblasts has significant implications for age-related skin wound healing. Chronic inflammation and increased ROS production contribute to delayed wound healing, impaired tissue regeneration, and increased susceptibility to chronic wounds in the elderly population [11,15]. Our findings show that the downregulation of miR-146a in aged fibroblasts is associated with enhanced inflammatory responses. Chronic inflammation is a hallmark of aging and contributes to delayed wound healing and impaired tissue regeneration [14,22]. Elevated levels of pro-inflammatory cytokines and chemokines in aged fibroblasts can lead to a prolonged inflammatory phase during wound healing, thereby impeding the healing process. Restoring miR-146a levels could help resolve chronic inflammation and promote a more efficient transition from the inflammatory to the proliferative phase of wound healing. The increased production of ROS in aged fibroblasts results in elevated oxidative stress, which can damage cellular components, including lipids, proteins, and DNA [17,20]. This oxidative damage contributes to cellular senescence and dysfunction. miR-146a’s regulation of NOX4 helps reduce ROS levels, thereby lowering oxidative stress and its detrimental effects. By enhancing antioxidant defenses, miR-146a can support the maintenance of cellular integrity and function in aged skin.

Moreover, the development of miR-146a-based therapeutics may have broader implications for the treatment of other age-related conditions characterized by chronic inflammation and oxidative stress, such as cardiovascular disease [27], neurodegenerative disorders [28], and metabolic syndrome [29]. Further research is warranted to elucidate the molecular mechanisms underlying the dysregulation of miR-146a expression in aged dermal fibroblasts and its impact on skin aging and wound healing. In addition, investigation into the therapeutic efficacy and safety of miR-146a-based interventions in preclinical models and clinical trials is essential for translating these findings into clinical practice. Moreover, exploring the potential synergistic effects of miR-146a with other therapeutic modalities, such as growth factors, cytokines, and stem cell therapy, may enhance its therapeutic potential and broaden its applicability in clinical settings.

In conclusion, our study provides novel insights into the role of miR-146a in modulating inflammatory responses and oxidative stress in aged dermal fibroblasts, highlighting its potential as a therapeutic target for addressing age-related skin wound-healing deficits. By targeting key pathways involved in inflammation resolution and antioxidant defense, miR-146a-based interventions hold promise for improving wound-healing outcomes and promoting skin health in the aging population.

## 4. Materials and Methods

### 4.1. Animal Studies

All animal experiments were approved by the Institutional Animal Care and Use Committee at the University of Tennessee Health Science Center, and experimental protocols followed the guidelines described in the NIH Guide for the Care and Use of Laboratory Animals. In these experiments, we used 17-week-old (young) and 88-week-old (aged) female C57BL/6J mice from the Jackson Laboratory (Bar Harbor, ME, USA). Mice were anesthetized with inhaled isoflurane. Each mouse was shaved and depilated before wounding. The dorsal skin was swabbed with alcohol and Betadine (Purdue Pharma, Stamford, CT, USA). Each mouse received a single, full-thickness dorsal wound (including panniculus carnosum) with an 8 mm punch biopsy (Miltex Inc, York, PA, USA). All wounds were dressed with Tegaderm (3M, St Paul, MN, USA), which was subsequently removed on postoperative day 2. Postoperatively, the mice received a subcutaneous injection of an analgesic, carprofen. Wound images were taken every other day and analyzed by ImageJ (https://imagej.nih.gov/ij/). A full-thickness skin sample, centered on the wound, was harvested 0 and 7 days after surgery (*n* = 5 per timepoint).

### 4.2. Dermal Fibroblast Isolation and Culture

Primary dermal fibroblasts were isolated from the skin of 17-week-old (young) and 88-week-old (aged) mice. They were cultured in a full medium comprising Dulbecco’s modified eagle high-glucose medium (DMEM, Sigma-Aldrich, St. Louis, MO, USA) supplemented with 10% fetal bovine serum (FBS; Gibco, MA, USA) and 1% antibiotic–antimycotic (Sigma-Aldrich, St. Louis, MO, USA) and maintained at 37 °C in a humidified atmosphere containing 5% CO_2_. For overexpression, fibroblasts were transfected with miR-146a mimics or control mimics. Transfection reagents, mimics, and control miRNAs were purchased from Invitrogen. Twenty-four hours following transfection, the cells were processed for analysis. The size of dermal fibroblasts of young and aged skin at passage 2 was analyzed and compared using the Echo Revole microscope (Echo, San Diego, CA, USA) and its built-in software.

### 4.3. Real-Time Quantitative PCR

Total RNA was extracted with TRIzol reagent (Invitrogen, Carlsbad, CA, USA) according to the manufacturer’s established protocol. Quantitative RT-PCR analyses for miR-146a and U6 (used as a normalization control) were performed using TaqMan miRNA assays with reagents, primers, and probes obtained from ThermoFisher Scientific. For gene expression analysis, RNA was converted into cDNA using the SuperScript First-Strand Synthesis System (Invitrogen, Life Technologies, Carlsbad, CA, USA). IRAK1, TRAF6, NF-kB, IL6, and NOX4 were amplified using the TaqMan gene expression assay (Applied Biosystems, ThermoFisher Scientific, Waltham, MA, USA). Internal normalization was achieved by using the GAPDH housekeeping gene. Samples (*n* = 5 per group) were amplified in triplicate and results were averaged for each individual sample. The ΔΔCT method was used to calculate relative gene expression. Results are reported as mean ± SD.

### 4.4. Intracellular ROS Measurement

After the fibroblasts (young or aged) reached 70% confluence, they were subjected to transfection with miR-146a mimics or control mimics. Intracellular production of hydroxyl, peroxyl, and other ROS was measured by the Cellular Reactive Oxygen Species Detection Assay Kit (Abcam, Waltham, MA, USA). After 24 h of transfection, the fibroblast cells were exposed to 2′,7′-dichlorofluorescein diacetate (DCFDA) for 20 min. The level of intracellular ROS was assessed by the fluorescence emitted by DCFDA after conversion to 2′,7′-dichlorofluorescein by reaction with ROS. The excitation and emission wavelengths were 492 and 521 nm, respectively; ROS levels were recorded in arbitrary units.

### 4.5. Statistical Analysis

Results are expressed as mean ± SD for *n* = 3 to 5 independent experiments. Statistically significant differences in gene expression between the two groups were assessed by Student’s *t*-test. *p* < 0.05 was considered to be statistically significant.

## Figures and Tables

**Figure 1 ijms-25-06821-f001:**
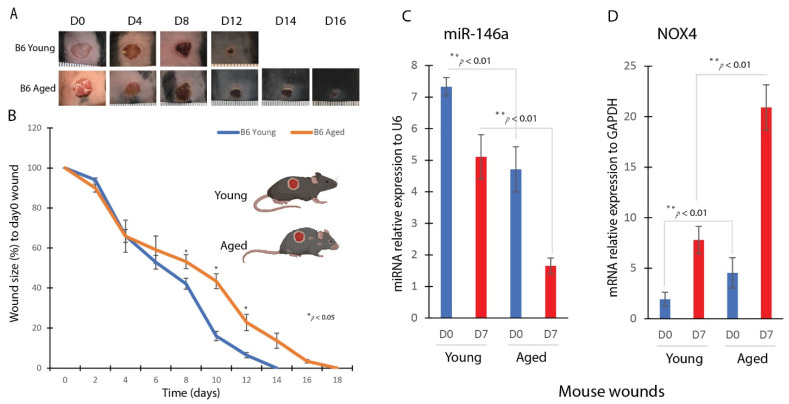
Delayed wound healing in aged mice with reduced miR-146a and induced NOX4 expression. (**A**) Representative images of wounds on days 0, 4,8, 12, 14, and 16 after wounding. (**B**) Wound size change during the healing process of initial 8 mm wound in WT (C57BL/6J or B6) mice at different ages by Image J analysis. Young (17 weeks old mice), blue; aged (88 weeks old mice), orange (*n* = 5 per group). Error bars indicate mean ± SEM. *, *p* < 0.05. (**C**) Real-time qPCR analysis of miR-146a gene expression in young and aged dermal fibroblasts (mean ± SD, *n* = 5 per group). (**D**) Real-time qPCR analysis of NOX4 gene expression in young and aged dermal fibroblasts (mean ± SD, *n* = 5 per group). ** *p* < 0.01.

**Figure 2 ijms-25-06821-f002:**
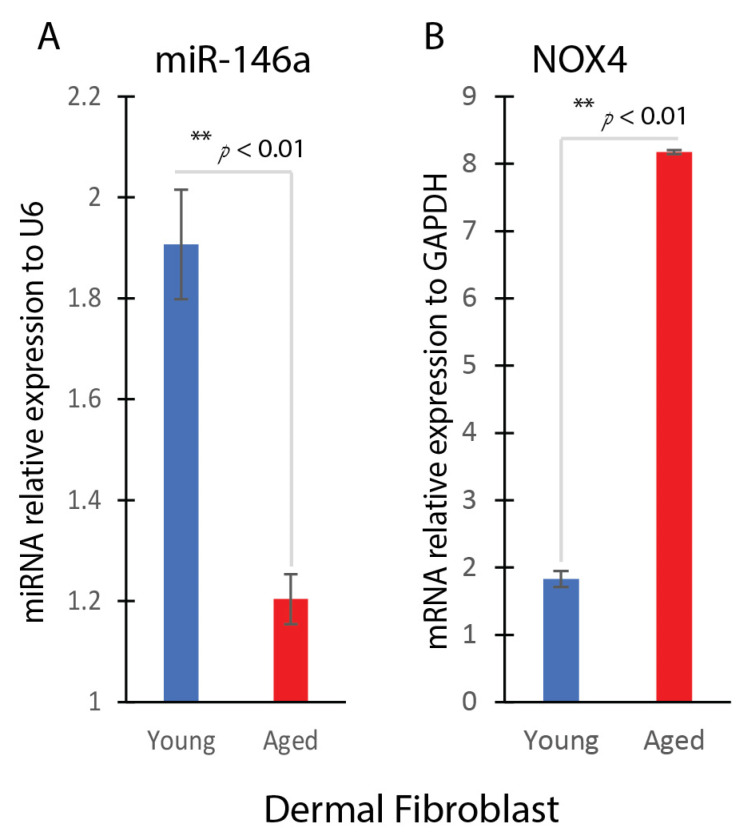
MiR-146a and NOX4 expression in aged dermal fibroblasts. (**A**) Real-time qPCR analysis of miR-146a gene expression in young and aged dermal fibroblasts (mean ± SD, *n* = 3 per group). (**B**) Real-time qPCR analysis of NOX4 gene expression in young and aged dermal fibroblasts (mean ± SD, *n* = 3 per group). ** *p* < 0.01.

**Figure 3 ijms-25-06821-f003:**
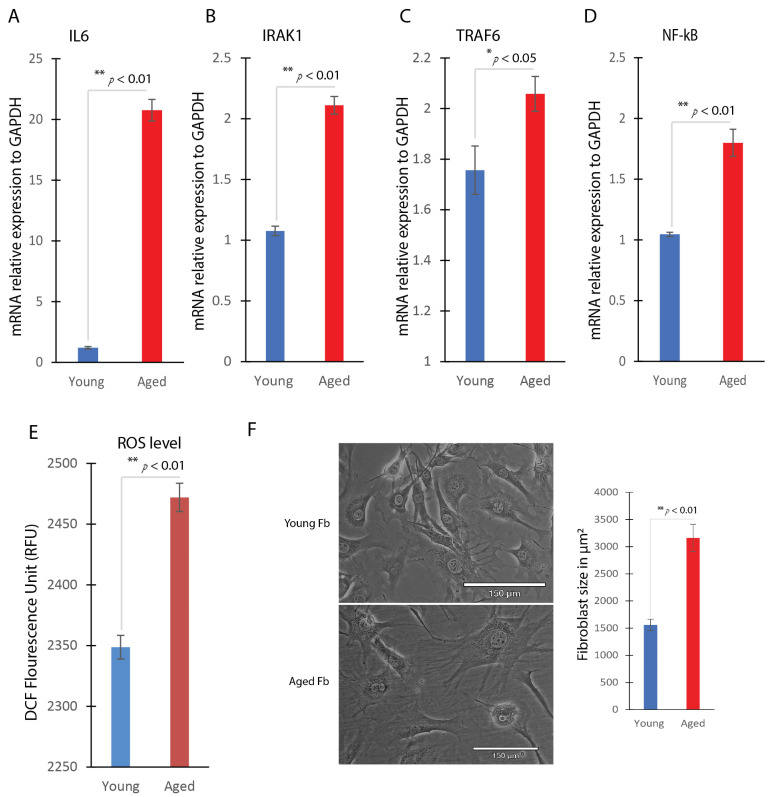
Inflammatory responses and ROS production in aged dermal fibroblasts. Real-time qPCR analysis of IL6 (**A**), IRAK1 (**B**), TRAF6 (**C**), and NF-kB (**D**) gene expression in young and aged dermal fibroblasts (mean ± SD, *n* = 3 per group). (**E**) ROS production between young and aged dermal fibroblasts using the DCFDA/H2DCFDA–cellular ROS assay. (**F**) Young and aged dermal fibroblast size comparison using the Echo Revolve macroscope’s built-in software (https://discover-echo.com/revolve). * *p* < 0.05, ** *p* < 0.01.

**Figure 4 ijms-25-06821-f004:**
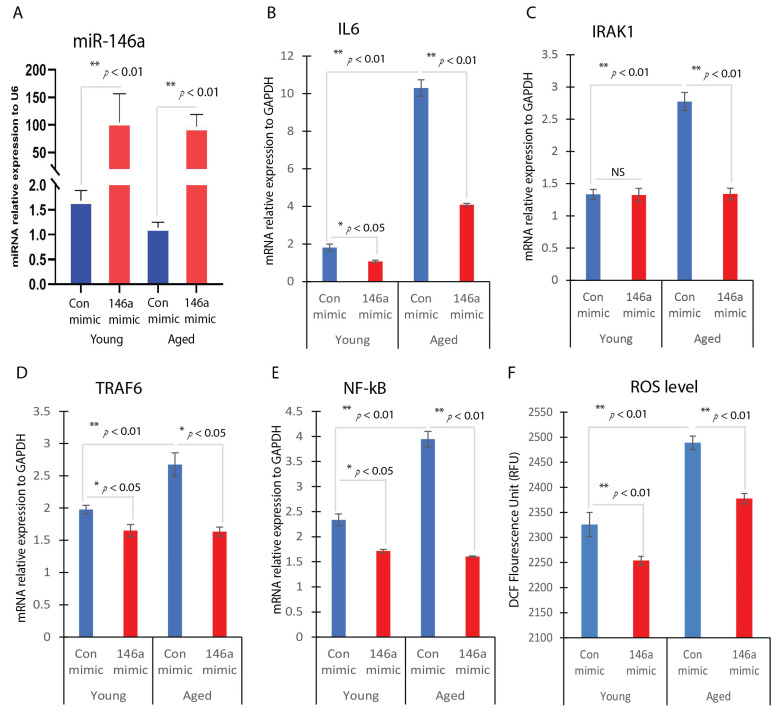
Effects of miR-146a overexpression on inflammatory responses and ROS production. Dermal fibroblasts were transfected with a miR-146a mimic (146a mimic) or a negative control mimic (Con-mimic). (**A**) miR-146a expression was detected by real-time qPCR with U6 as an internal control. The gene expression levels of IL6 (**B**), IRAK1 (**C**), TRAF6 (**D**), and NF-kB (**E**) were determined by RT-qPCR in miR-146a overexpression fibroblasts or control mimic-transfected fibroblasts. (**F**) ROS production between young and aged dermal fibroblasts transfected with miR-146a mimics or control mimics using the DCFDA/H2DCFDA–cellular ROS assay.

**Figure 5 ijms-25-06821-f005:**
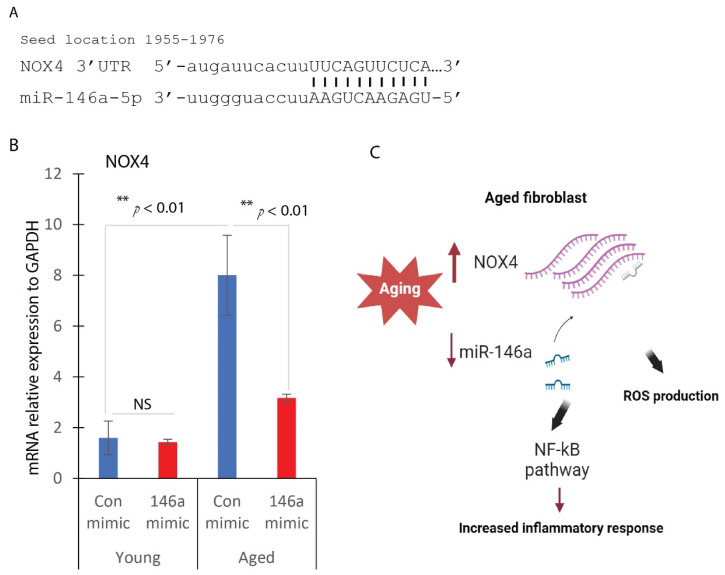
MiR-146a targets NOX4 in dermal fibroblasts. (**A**) Bioinformatic analysis of miRNA targets revealed a binding site between miR-146a and the 3′-UTR region of human Nox4 mRNA, suggesting a possible regulatory relationship between miR-146a and Nox4. (**B**) NOX4 gene expression in miR-146a overexpression young or aged fibroblasts (mean ± SD, *n* = 3 per group). ** *p* < 0.01. NS: not significant. (**C**) Cartoon illustration of possible mechanisms of miR-146a/NOX4 signaling in aged dermal fibroblasts.

## Data Availability

Data is contained within the article.

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
