# Peer review of "miR-146a Decreases Inflammation and ROS Production in Aged Dermal Fibroblasts"

_ijms, 2024, doi:10.3390/ijms25136821_

Round 1

Reviewer 1 Report

Comments and Suggestions for Authors

Thank you for the opportunity to review this manuscript by Zhang and colleagues.  These studies explored the role of miR-146a in dermal fibroblast senescence.  This is an important topic as dermal senescence plays important roles in aging-related disorders such as wound healing and non-melanoma skin cancer.

The authors show decreased wound healing in aged mice, which correlates with miR-146a and NOX4 expression levels.  Next, murine fibroblasts were obtained from young and old mice with similar features.  Cytokine levels were elevated in the aged fibroblasts along with decreased MiR-146a levels—overexpression of this miRNA mimic inhibited the cytokine/ROS responses.  The studies are overall well-organized and this work has merit.  Would have been interesting to use agomirs/antagomirs in the murine wounding studies but, as the differences were not overly dramatic, these would be challenging to interpret. 

The manuscript is well-written.

My only suggestions are relatively minor-

1) Consider making the title plural for fibroblast

2)  the authors should consider including a recent manuscript that examined miR-146a and human dermal fibroblast aging.  doi: 10.3389/fphys.2024.1291344. PMID: 38487265; PMCID: PMC10937357

Author Response

Comment 1: Thank you for the opportunity to review this manuscript by Zhang and colleagues. These studies explored the role of miR-146a in dermal fibroblast senescence. This is an important topic as dermal senescence plays important roles in aging-related disorders such as wound healing and non-melanoma skin cancer.

Response: We appreciate the positive feedback and recognition of the importance of our research topic.

Comment 2: The authors show decreased wound healing in aged mice, which correlates with miR-146a and NOX4 expression levels. Next, murine fibroblasts were obtained from young and old mice with similar features. Cytokine levels were elevated in the aged fibroblasts along with decreased miR-146a levels—overexpression of this miRNA mimic inhibited the cytokine/ROS responses. The studies are overall well-organized and this work has merit. Would have been interesting to use agomirs/antagomirs in the murine wounding studies but, as the differences were not overly dramatic, these would be challenging to interpret.

Response: We are grateful for the recognition of the organization and merit of our work. We agree that the use of agomirs/antagomirs could provide further insights, though their interpretation could indeed be challenging given the subtle differences observed. We will consider this approach for future studies to enhance the understanding of miR-146a's role in wound healing.

Comment 3: The manuscript is well-written. My only suggestions are relatively minor:

Consider making the title plural for "fibroblast".

The authors should consider including a recent manuscript that examined miR-146a and human dermal fibroblast aging. doi: 10.3389/fphys.2024.1291344. PMID: 38487265; PMCID: PMC10937357

Response: We appreciate the suggestions and will implement the following changes:

We will revise the title to "MiR-146a decreases inflammation and ROS production in aged dermal fibroblasts."

We will include the recommended reference (doi: 10.3389/fphys.2024.1291344. PMID: 38487265; PMCID: PMC10937357) in the manuscript to provide a comprehensive overview of miR-146a's role in human dermal fibroblast aging.

Reviewer 2 Report

Comments and Suggestions for Authors

Review report for the manuscript entitled „MiR-146a decreases inflammation and ROS production in aged dermal fibroblast“ by Liping Zhang, Iris C. Wang, Songmei Meng and Junwang Xu.

In this study the ageing skin and functionality of dermal fibroblasts was investigated with a special focus on miRNA146a to compensate the impairments. The authors associate the decline in functionality of dermal fibroblasts during skin ageing with delayed wound healing and demonstrate this in a wound healing model in young and aged mice. As a critical regulator of inflammation and oxidative stress responses miRNA 146a was shown to be significantly reduced in expression in ageing fibroblasts of wild type mice. Its overexpression in young and aged dermal fibroblasts attenuates the inflammatory response and reduced the ROS levels. The study was performed in young (17 weeks) and aged (80 weeks) mice and on isolated cultured fibroblasts. Naturally miRNA 146a expression declines with ageing as shown for fibroblasts of wild type mice while its overexpression could offer a therapeutic strategy for combating age related wound healing deficits as the authors suggest.

The study is very interesting and may hold potential for therapeutic applications since the effects on inflammatory cytokines and NFkB pathway are pronounced in aged fibroblasts which may have implications in other related metabolic pathways or diseases. However, the mansucript contains some flaws that need the authors attention and revision.

There are only 19 references in the entire manuscript. 16 of them are cited in the introduction. The discussion contains very few (only two) references. The proportion is inacceptable. The authors should revise the discussion carefully and include further references into the discussion to put their findings in the context of current research. Currentlx it is written very general. Moreover, the introduction is very short on the specific pathways that were investigated. The author shoud improve here and add more content to this end.

Some specific points follow:

-        Regarding the animal study: the age of ageing animals is inconsistent. It is given with 80 weeks (line 17) and 88 weeks (line 77). Please check and correct.

-        The culture medium contained only FBS 10% but no other supplements such as antibiotics, glutamine or others. Is that correct?

-        Within the materials and methods, the information on city, state and country are missing in some cases, i.e. quiagen, invitrogen, Applied Biosystems …Please provide these information.

-        Please use the same units throughout the manuscript (hrs or h, line 95 vs. 113)

-        What are the RAW cells used for (line 113)? There is no further explanation to it. Please add.

-        What was used as controls? Line 100: RNU6 was used as a normalization control. In Fig. 2 the miRNA relative expression was related to U6. Are these two the same? Please use a uniform label and describe the controls clearly.

-        The sample size of 3-5 is very small for a meanigful statistical analysis by t-test.

-        Fig 1 : Wound images are labelled with B6. What does it stand for? Please use uniform names throughout the manuscript.

-         

-        In parts interpretation of the results is included in the results section, for instance „…NOX4, the primary enzyme responsible for generating ROS in fibroblasts [17, 18], within the aged fibroblasts.“ This should be separated and references should not be used when describing the results unless necessary.

-        For Fig2 it should be clearly stated that here cultured fibroblast were use in contrast to Fig 1 where wound tissue samples were used.

-        ROS assay: Difference in fluorescence intensity is a bit more than 100 RFU. In Fig 4F the results regarding ROS levels are described as % decrease. In the diagram the units are shown as RFU. For a better comparison it would be helpful if the same relation could be applied to the results in Fig 3E.

-        In Fig 3 the size of fibroblasts of young and aged cells is compared by using Echo Revolve macroscope built-in software as is written in the legend text. Please describe the method in chapter 2 material and methods and give some information on the microscope and software (version, company) and days of culture of cells at the time of measurement.

-        In Fig 3 the size of diagramms is different. Please adjust and add a scale bar to the microscopic image. The image appears very dark. Maybe the quality can be improved.

-        Fig4a) The legend text reads that the miR-146a expression was related to U6 as internal control. However, the diagramm has a different axis labelling. Please check and correct.

-        Please check the numbering in Fig 4.F with legend text (F does not appear).

-         

-        The discussion is written very general but does not refer to the authors own findings. Furthermore, it contains only two references. The authors are encouraged to specifically discuss their results in the context of current literature within the discussion.

Comments on the Quality of English Language

English language is fine.

Author Response

Comment 1: Review report for the manuscript entitled „MiR-146a decreases inflammation and ROS production in aged dermal fibroblast“ by Liping Zhang, Iris C. Wang, Songmei Meng and Junwang Xu.

In this study, the ageing skin and functionality of dermal fibroblasts were investigated with a special focus on miRNA146a to compensate for the impairments. The authors associate the decline in functionality of dermal fibroblasts during skin ageing with delayed wound healing and demonstrate this in a wound healing model in young and aged mice. As a critical regulator of inflammation and oxidative stress responses, miRNA 146a was shown to be significantly reduced in expression in ageing fibroblasts of wild-type mice. Its overexpression in young and aged dermal fibroblasts attenuates the inflammatory response and reduces ROS levels. The study was performed in young (17 weeks) and aged (80 weeks) mice and on isolated cultured fibroblasts. Naturally, miRNA 146a expression declines with ageing as shown for fibroblasts of wild-type mice while its overexpression could offer a therapeutic strategy for combating age-related wound healing deficits as the authors suggest.

Response: We appreciate the detailed summary and positive feedback on our study.

Comment 2: The study is very interesting and may hold potential for therapeutic applications since the effects on inflammatory cytokines and NFkB pathway are pronounced in aged fibroblasts which may have implications in other related metabolic pathways or diseases. However, the manuscript contains some flaws that need the authors' attention and revision.

Response: We are grateful for the recognition of the potential therapeutic applications of our study. We addressed the noted flaws.

Comment 3: There are only 19 references in the entire manuscript. Sixteen of them are cited in the introduction. The discussion contains very few (only two) references. The proportion is unacceptable. The authors should revise the discussion carefully and include further references to put their findings in the context of current research. Currently, it is written very generally. Moreover, the introduction is very short on the specific pathways that were investigated. The authors should improve here and add more content to this end.

Response: We acknowledge the need for a more thorough literature review and contextualization of our findings. We revise the discussion to include additional relevant references, ensuring a comprehensive review of current research in the field. We also expand the introduction to provide more detailed information on the specific pathways investigated, such as the NF-κB pathway and NOX4, to offer a clearer understanding of the study's context.

Comment 4: Some specific points follow

Response: We thank the reviewers for identifying typographical errors and other issues in our manuscript. We have addressed each point according to their comments and suggestions. The revisions will enhance the manuscript's depth and clarity, ensuring it provides a comprehensive and well-supported exploration of miR-146a's role in dermal fibroblast aging and potential therapeutic applications.

Round 2

Reviewer 2 Report

Comments and Suggestions for Authors

The authors have revised the manuscript and considered the recommendations.